# Establishing a dementia care competency framework for care partners, health and social care providers: A modified Delphi study protocol

Kelly Kay[1], Kateryna Metersky[2], Rebecca H. Correia[3,4]*, Arlene Astell[5,6,7], Colleen McGrath[8], Winnie Sun[9], Halyna Yurkiv[10], Victoria Smye[10]

1 Provincial Geriatrics Leadership Ontario, Toronto, Canada, 2 Daphne Cockwell School of Nursing, Toronto Metropolitan University, Toronto, Canada, 3 Department of Health Research Methods, Evidence, and Impact, McMaster University, Hamilton, Canada, 4 Department of Family Medicine, Dalhousie University, Halifax, Canada, 5 Department of Occupational Sciences and Occupational Therapy, University of Toronto, Toronto, Canada, 6 Dementia Ageing Technology Engagement Lab, University Health Network, Toronto, Canada, 7 School of Psychology, Northumbria University, United Kingdom, 8 School of Occupational Therapy, Western University, London, Canada, 9 Faculty of Health Sciences, Ontario Tech University, Oshawa, Canada, 10 Arthur Labatt Family School of Nursing, Western University, London, Canada

* correirh@mcmaster.ca

## Abstract

Dementia care requires a wide range of knowledge and skills delivered by both unpaid care partners and health and social care providers. In Ontario, Canada, no systematic framework currently aligns educational content with dementia care competencies. This gap risks the effectiveness of dementia-related education and care delivery. Therefore, this study protocol describes our approach to achieve consensus on the behavioural statements that describe the core competencies required of care partners, health and social care providers involved in dementia care. We will use a two-round modified Delphi method with expert panellists from two groups: (1) care partners with experience caring for someone living with dementia and (2) interprofessional health and social care providers working with people living with dementia. We will purposively recruit up to 80 panellists (40 per group). Panellists will assess standardized behavioural competency statements derived from earlier study phases, rating them on importance and measurability using a nine-point Likert scale. Round 1 will include opportunities for panellists to suggest new statements. Statements reaching ≥70% agreement (rated 7–9 on a 9-point Likert scale) and demonstrating a narrow interquartile range (IQR ≤ 2) will advance to Round 2. In the second round, a higher consensus threshold (≥80%) and stability in ratings (median shift ≤1 point) will determine final inclusion. Qualitative feedback through open-ended questions will be analyzed alongside quantitative results to refine the statements. Findings will support the development of a consensus-based Dementia Care Competency Framework to guide evidence-based educational initiatives and care delivery across settings. This inclusive approach will provide a model for ensuring

**Data availability statement:** Anonymized survey data collected as part of the modified Delphi procedure can be made available upon reasonable request by contacting the corresponding author.

**Funding:** The author(s) received no specific funding for this work.

**Competing interests:** The authors have declared that no competing interests exist.

both lived experience and clinical expertise shape the future of dementia care education in Canada and beyond.

## Introduction

Dementia is an increasingly prevalent chronic condition globally [1,2], and presents complex challenges for those who provide dementia care [3]. Dementia care involves a broad range of knowledge, skills, and sound judgment to enable actions that effectively support individuals living with dementia, often requiring substantial emotional, physical, and cognitive effort [4]. Dementia care is not limited to clinical settings or regulated health professionals; it is delivered in partnership with interprofessional health and social care providers and care partners (also termed informal or unpaid caregivers [5]), such as family members or friends [6]. Public education can support care partners of people living with dementia to better understand symptoms and approaches to care, and to encourage their pursuit of evidence-based information and timely assistance [7]. Given the complexity of dementia care and the wide range of health and social care providers involved, there is a critical need to identify and define the core competencies necessary to ensure safe, person-centered, and effective support and care.

Competency frameworks are used extensively in health care settings to coordinate professional education and provide a systematic approach to the development, delivery, and evaluation of curricula [8,9]. Competencies express behaviours that can be translated across multiple roles, activities, and tasks by applying core knowledge, skills, attitudes, judgements, or actions [10]. Competency-based dementia education may assist carers by translating emerging dementia research into both lay and professional dementia care activities and identifying specific desired outcomes for learning [11,12]. In Canada, a proliferation of national and provincial education and training modules have attempted to address knowledge gaps in dementia care for a broad range of audiences [13]. However, to date there has been no systematic approach mapping educational content to dementia care competencies and none including competencies specific to the identified wishes and needs of people living with dementia and their care partners. This gap may impede knowledge translation and evaluation efforts, risking educational training devoid of evidence-based competencies [14].

Building on a dementia care competency framework developed in the United Kingdom [15] and extensive input from public consultations with people who live with dementia and their care partners across Ontario, Canada, this study will develop a competency framework to guide approaches to evidence-based education related to dementia care for two audiences: (i) care partners with lived experience and (ii) health and social care providers. The purpose of this study is to reach consensus among people with lived experience, including care partners, health and social care providers, about the behavioural statements that describe the core competencies required of all individuals caring for people living with dementia. This study will answer the following research question: *Which competency statements describe the*

*roles, responsibilities, and expectations of care partners and interprofessional health and social care providers who deliver service and care to people living with dementia?* We hypothesize that a core subset of competencies aligning with person-centered dementia care will achieve consensus. We anticipate that there will be some divergence in the competencies prioritized by care partners compared to health and social care providers.

## Materials and methods

### Design

We will conduct a modified Delphi study to achieve consensus on the core competencies that will inform the development of a Dementia Care Competency Framework. Consensus methods, such as the Delphi approach, systematically measure and establish agreement through a series of structured consultations with a technical expert panel [16]. These methods are based on the premise that an accurate and reliable assessment can be achieved by consulting a panel of experts and developing group consensus [16,17]. A modified (rather than a traditional) Delphi design was chosen as pre-defined competency statements developed in earlier study phases will be evaluated during the consensus-building process. Our study conceptualization and methodological planning aligns with the ACcurate COnsensus Reporting Document (ACCORD) (S1 Table) [18].

### Setting

Although the consensus-based competencies may be transferrable across geographic settings, this work will be carried out in the province of Ontario, Canada. As in most of the world, dementia care in Ontario is delivered largely by care partners (e.g., family members, friends). Across the province, additional support may be available from home care, community care, social service agencies, and clinical services, including primary care and specialized geriatric services.

In 2017, the Ontario Ministry of Health and Long-Term Care announced a 10-pillar provincial dementia strategy, identifying dementia workforce and care partner training and education as two key investments. Learning needs related to the provision of dementia care were ranked among the top priorities identified in a meta-summary of training needs assessments of Ontario health professionals [19]. Further, approximately half of attendees at an annual specialized geriatric services educational event identified the need for skills development related to dementia care. One-third of attendees identified seniors' mental health and dementia care as a perceived practice gap among non-specialized geriatric service providers who work with older adults [20].

### Technical expert panel

We will engage panellists from two distinct groups, with the following inclusion criteria: (i) Care partners – People with lived experience as care partners providing unpaid care to people living with dementia; and (ii) Interprofessional health and social care providers – Individuals who work with people living with dementia and their care partners in a professional (paid) capacity. Health and social care providers must have a minimum of three years of experience working directly with people living with dementia, be a member of a regulated or unregulated health or social care profession that provides services to people living with dementia, and hold current employment in an organization providing dementia care or a closely related service. We aim to recruit diverse health and social care providers, including personal support workers, emergency medical service workers, counsellors, nurses, physicians, physiotherapists, occupational therapists, social workers, dietitians, and speech language pathologists, among others from across Ontario, Canada.

### Recruitment and sample size

We will employ purposive/criterion sampling to recruit up to 80 panellists (40 per group), which aligns with the recommended sample size to not overload, demotivate, or disengage participants [21]. Given limited guidance on the target

sample size for modified Delphi studies, we aim to strike a balance between gathering input from more panellists with a large, representative sample and the potential for continuous dissensus [21,22]. We will identify prospective panellists through local Alzheimer Societies, Regional Geriatric Programs, professional associations, aging research networks/institutes, dementia care organizations, and the Provincial Geriatrics Leadership Ontario network. Panellists will be recruited via email to describe the study, the scope of their engagement, the time commitment, and how their responses will be applied in the study [23]. For those willing to participate, we will provide a link to the Letter of Information and Consent to learn more about the opportunity. After reviewing the study details and providing written consent, panellists will complete a demographic survey to confirm their eligibility and inform our ongoing recruitment efforts with the goal of achieving maximum variation sampling (e.g., to ensure gender balance, representation across health and social care professions, residence across Ontario, etc.). Panellists will be expected to participate in both study rounds to ensure ongoing engagement in consensus-building [24]. Following each round, personal emails will be sent to panellists thanking them for their contributions and advising on next steps (e.g., timeline of the analysis period, instructions for the second survey) [25].

## Delphi survey development

In each Delphi survey, panellists will assess a series of behavioural statements aligned with the vetted principles for dementia care (Table 1) [13]. The proposed behavioural statements were developed using multiple approaches from earlier study phases. Behavioural statements originally described by Bardsley (2011) were adapted based on feedback received during a series of consultations with people living with dementia, care partners, and clinicians in 2013 and 2014 [15]. These consultations, led by KK, VS, and AA, resulted in an initial working set of behavioural statements. The statements were further revised based on integrating reflections and suggestions raised by focus group participants in 2023 and 2024 with people living with dementia and their care partners. Lastly, experts in competency development, adult education, dementia care, and aging research contributed to refining the behavioural statements to remove duplication and align them with competency-based education theory from 2024 to 2025. Proposed statements for the Delphi survey were standardized to: (i) clearly describe what individuals providing dementia care ought to be able to know and do; (ii) include a single, identifiable, observable, and measurable action or behaviour; (iii) ensure a set of "mutually-exclusive, non-overlapping actions or behaviour" [26]; and (iv) follow a consistent format of what (action required), who/what (subject/object involved), why (purpose), and how (approach) [26].

The research team will closely review behavioural statements in multiple stages to ensure accurate wording, thereby reducing bias and response variance [27,28]. Prior to disseminating the Delphi surveys, our team will pilot test them to ensure comprehension, readability, and functionality [27].

Table 1. Examples of Behavioural Statements to be Rated in the Delphi Survey.

| Subcategory | Competency Statement |
| --- | --- |
| **Principle 1: Promote Health and Social Wellbeing** | |
| Dementia Awareness and Advocacy | Demonstrates expert and detailed knowledge of dementia prevention and related information (e.g., genetics) to inform care practices by staying current with scientific research and guidelines. |
| **Principle 5: Communicate Sensitively/Safely** | |
| Communication and Interaction | Tailors communication with people living with dementia to match shifting cognitive and behavioural capabilities and communication needs by using tools such as handouts, speaking slowly and repeating messages, adjusting tone and pitch, and employing body language cues. |
| **Principle 9: Work as part of a Multi-Agency Team to Provide Support** | |
| System Navigation and Resource Awareness | Applies knowledge of the roles and services of health and social service organizations supporting people living with dementia to guide referrals and care planning, by identifying appropriate services and explaining their functions. |

## Overview of the Delphi rounds

We will facilitate two Delphi rounds given the consistent purpose of both surveys and an increasingly strict threshold for consensus over successive rounds to establish a core competency set (Fig 1) [22,29]. In Round 1, panellists will rate the proposed competency statements and have the opportunity to suggest additional statements for consideration in Round 2. In the second round, we aim to achieve agreement on the core competencies with a higher threshold for consensus.

## Data collection and analysis

In the first Delphi round, an online questionnaire will be developed and distributed to panellists to complete within two weeks. Study documents (i.e., this study protocol, letter of information, consent form, and questionnaire instructions) will be emailed to the panellists, along with a link to access the survey via Microsoft Forms [30]. The intent of this round is to achieve agreement on the core competency statements identified in earlier study phases against our evaluative criteria, and suggest additional competencies that panellists perceive as missing [16]. In addition to providing numeric ratings on our set evaluation criteria, panellists will have the opportunity to submit open-ended feedback to revise the competency descriptions or rationalize their ratings.

Between the first and second rounds, we will combine the judgements of panellists using statistical integration for numeric ratings and content analysis of open-ended responses. For each survey item (competency statement), we will report the median, interquartile range (IQR), and percentage of agreement to discern consensus and quantify its degree [31]. Medians are considered well-suited for ordinal data and to reflect convergence of opinion [32]. We will stratify and total the survey responses between the two groups of panellists (i.e., care partners versus health and social care providers) to examine differences and similarities in the prioritized statements. Individual feedback will be shared with each

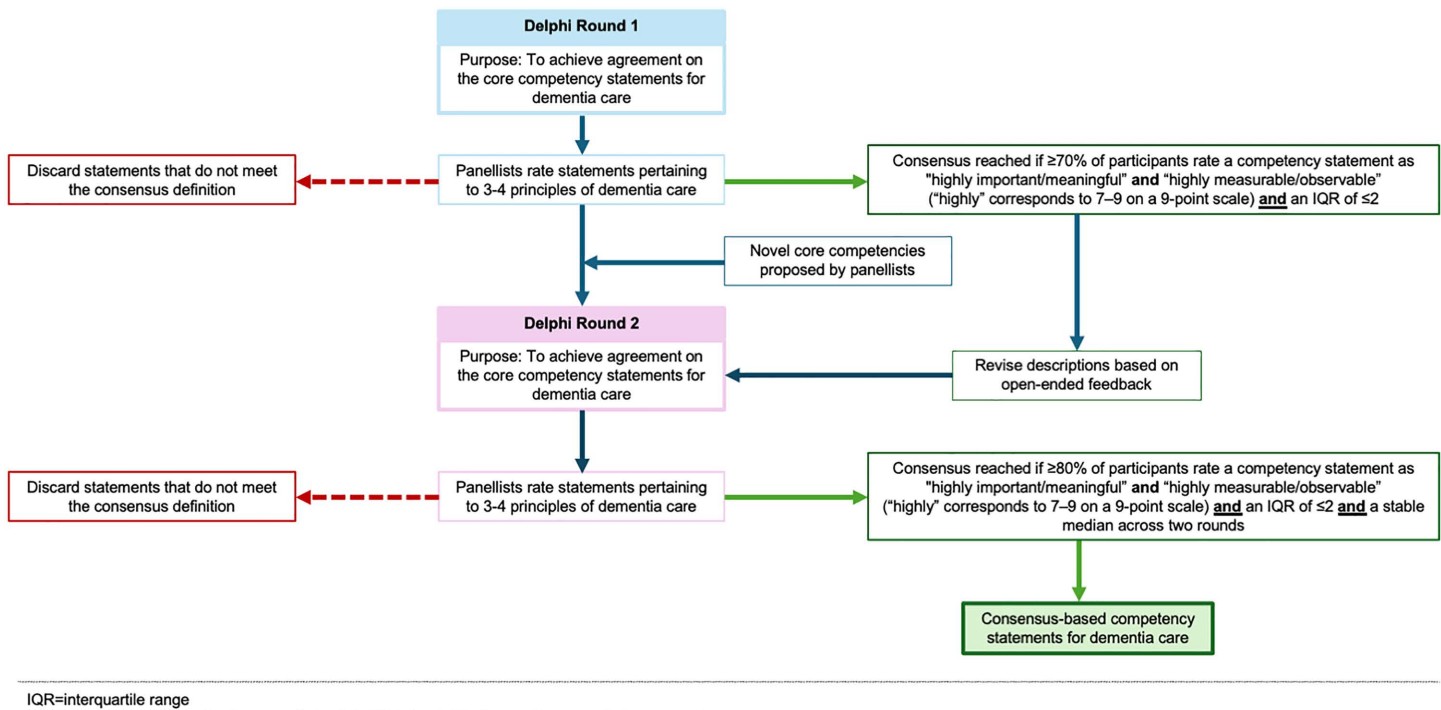

IQR=interquartile range
* Stable median will be judged as no substantial shift (<1 point) in the median score between rounds

**Fig 1. Approach to consensus.**

panellist before proceeding to the second survey. Panellists will be able to understand their ratings relative to the group/panel and consider new competencies identified by other panellists.

The second Delphi survey will be modified from the original survey based on competencies that achieved consensus or were suggested as novel by panellists. The same analysis approach as in Round 1 will occur following the second Delphi survey. In addition, we will compute a post-hoc Mann-Whitney U test to check for non-response bias [27]. We will also conduct the Wilcoxon matched-pairs signed-rank test to measure changes in consensus between rounds [33].

## Measurement

Numeric ratings and free-text responses will be collected in the online surveys. Given the large volume of competency statements to be assessed in this study (n = 281), panellists will be assigned to three or four principles of dementia care out of 14 total (organized into "work packages"), and will only evaluate the associated behavioural statements within those principles (Table 2) [13]. Given guidance on the number of Delphi items to not overwhelm or exhaust panellists [22,34], this approach is justified based on the large number of evidence-informed statements entering the Delphi for rating.

In the Delphi surveys, panellists will assign ratings on two criteria:

- **Importance/meaningfulness** (How essential is this competency statement to dementia care? How important is it to people living with dementia and their care partners?)

- **Measurable/observable** (How feasible is it to measure or observe this competency?)

We will collect ordinal ratings using a nine-point Likert scale, which aligns with methodological guidance and allows for granular measurement [35]. Panellists will have the opportunity to respond "cannot rate/answer" if they do not understand or feel they cannot assess a particular statement. Panellists will also indicate whether the competency statement pertains to (i.e., should be exhibited by) care partners, health and social care providers, or both groups. We will also pose open-ended questions in the questionnaire to solicit insights and other feedback: [27,32]

**Table 2. Principles of Dementia Care to Organize Behavioural Statements in the Delphi Surveys.**

| Work Package 1 | Principle 1 | Promote health and social well-being | 19 behavioural statements | 94 behavioural statements |
|---|---|---|---|---|
| | Principle 2 | Communicate sensitively and safely | 21 behavioural statements | |
| | Principle 3 | Identify dementia – know the early signs | 10 behavioural statements | |
| | Principle 4 | Screen, assess and diagnose | 28 behavioural statements | |
| | Principle 5 | Embed quality improvement, accountability and evaluation in dementia care practice | 16 behavioural statements | |
| Work Package 2 | Principle 6 | Care planning, treatment, intervention and follow-up | 25 behavioural statements | 93 behavioural statements |
| | Principle 7 | Facilitates living well with dementia; Promotes independence and encourages activity | 39 behavioural statements | |
| | Principle 8 | Understand the context of care and support for persons living with dementia and their care partners | 15 behavioural statements | |
| | Principle 9 | Provide palliative and end-of-life care for persons living with dementia | 14 behavioural statements | |
| Work Package 3 | Principle 10 | Understand and respond to unmet needs and signs of distress | 23 behavioural statements | 94 behavioural statements |
| | Principle 11 | Value and respect family and other care partners; Support access to services | 26 behavioural statements | |
| | Principle 12 | Work as part of a multi-agency team to provide support | 11 behavioural statements | |
| | Principle 13 | Support dementia worker personal development and self-care | 16 behavioural statements | |
| | Principle 14 | Clarify the accountability of leaders for processes and practices in dementia care | 18 behavioural statements | |

- **Comment box 1:** Please provide any feedback or revisions to the descriptions of competency statements.

- **Comment box 2:** Please provide any comments to explain or justify your ratings.

- **Comment box 3** (appears in Round 1 only): Please provide any suggestions of additional competencies associated with this principle that you feel are missing from those you evaluated in this section.

Surveys will collect ratings from panellists over a two-week period and will take approximately 60–90 minutes to complete.

## Assessing consensus

Given diverse approaches to define consensus [36], we elected to establish a robust consensus definition using multiple combined criteria. In the first round, we will measure the proportion of agreement for each competency within a pre-defined range and assess the spread of responses [31]. Competency statements will be retained if (i) at least 70% of panellists rate the indicator between 7–9 on our 9-point Likert scale for "important/meaningful" and "measurable/observable" and (ii) a narrow IQR is achieved (i.e., IQR ≤ 2), indicating strong agreement. Competency statements that do not meet this threshold for both criteria will be eliminated from subsequent rounds. In the second round, endorsed competencies will move into our final set if (i) the same criteria as the first round is met with an increased threshold of 80% and (ii) stability across both rounds is exhibited. Stability will be determined if there is no substantial shift (>1 point) in the median score of an individual competency statement between rounds.

## Ethical considerations

We obtained approval from the institutional Research Ethics Board (project #111589) at the University of Western Ontario. Prospective panellists will review and sign a letter of information and consent before participating in this study. At the start of each survey, panellists will affirm their informed consent to participate in the Delphi procedure, in writing, and will be reminded that their participation is voluntary. Panellists can decline to answer any survey questions and can withdraw their participation at any time.

## Data management

The survey will be developed and administered using Microsoft Forms. Anonymized survey results will be stored in a secure OneDrive folder accessible only to members of the research team. Each panellist will be assigned a unique identification number that will be entered at the start of each survey to track their participation across study rounds. Only two investigators (KK and RHC) will have access to the study key linking the names of panellists with their identification number, which will be stored in an encrypted, password-protected spreadsheet. Therefore, all data collected in this study will be anonymized; no identifiable information will be available when analyzing the survey data.

## Status and timeline

The conceptualization and planning of this Delphi study began in May 2025. Identification of prospective panellists is underway, with recruitment expected to begin in September 2025. The first Delphi survey will be pilot tested in late September 2025 and will launch in October 2025. Following the first Delphi round, investigators will analyze the survey findings and develop the second Delphi survey based on the initial findings (i.e., removing statements that do not achieve consensus, adding statements suggested by panellists, revising statements to improve comprehension based on panellist feedback). Following this analysis period, the second Delphi survey will launch, and data will be collected over a two-week period. We anticipate this study will be completed by December 2025.

## Discussion

### Candidate competency statements for rating

The adaptation of behavioural statements [15] and integration of concepts from lived experience advisors and experts in education, clinical care and research resulted in 428 candidate statements. Across these original statements, there was inconsistent application of the "what, who/what, why and how" format and significant duplication. The lead investigator (KK) reviewed each statement to ensure alignment with the original intent, revised them according to the standard format, and suggested sub-categories aligned with the vetted principles for dementia care. The research team met in-person in July 2025 to refine the resultant 367 candidate statements and further identify remaining duplication (i.e., exact replication or highly similar wording or intent). Ultimately, this process reduced the number of competency statements to 281 for rating in the first Delphi survey.

### Patient and public engagement

This study integrates an emancipatory paradigm and experience-based design to include people with lived experience directly in research activities [37,38]. This paradigm supports the position of the researchers that including care partners of people living with dementia is foundational to driving what is taught and learned about care needs. In keeping with these theories, earlier study phases that informed the development of candidate behavioural statements (to be evaluated in the Delphi procedure) engaged people living with dementia and their care partners in focus groups and consultations. Further, people with lived experience (both care partners and health and social care providers) will be consulted in the consensus-building activities through the Delphi process.

### Strengths and limitations

This study will engage diverse members of the lay public and health and social care providers from across Ontario (with different life experiences, training/education backgrounds, geography, sex, race, ethnicity) who will bring varied perspectives to this panel. The proposed competency statements were informed by published literature and the perspectives of people with lived experience, and exhibited breadth across the previously identified principles of dementia care established by the research team [13]. However, the large volume of statements (n = 281) cannot be feasibly assessed by every panellist, given concerns about participant burden and survey fatigue, resulting in the decision to assign panellists to rate a subset of competency statements (n = 93 or n = 94; see Table 2). Efforts will be made to balance the characteristics, experiences, and qualifications of those assigned to each work package based on responses to the demographic survey (e.g., the proportion of care partners to health and social care providers; disciplines of providers). We will maintain regular communication with panellists over the study period (such as sharing a summary of ratings after each survey round) to reduce the risk of attrition. We will gather and analyze both numeric ratings and open-ended responses, offering detailed explanations and clarifications on the proposed statements. Lastly, by publishing this study protocol, we have established an audit trail to maintain rigor in our survey development, analysis, and reporting of results.

### Dissemination plans

We will publish the resulting consensus on the core competency statements that describe the expectations of care partners and interprofessional health and social care providers in caring for people living with dementia. We plan to disseminate the study findings widely at formal (e.g., research conferences) and informal (e.g., internal organizational meetings) scientific and public events. In addition, we plan to share our findings with Ontario-based academic institutions that prepare health and social care providers to embed key learnings into their educational programs, in order to enhance future care delivery. The findings should also be of interest to education providers across Canada and globally. The consensus-based behavioural statements established in this Delphi study will directly inform the development of a Dementia Care

Competency Framework. There are several intended use cases for this Framework: (i) To support people living with dementia and their care partners to participate in and contribute to their communities for as long as they are able; (ii) To deliver high-quality person-centred service and health and social care, whenever people with dementia are interacting with those in any type of service organization; (iii) To identify training needs and facilitate the design and delivery of appropriate health and social care education and professional development, in response to the needs of people with dementia and their families; and (iv) To create role profiles, position descriptions, and performance frameworks that support quality assurance and excellence in dementia care.

## Supporting information

**S1 Table. Completed reporting checklist.**
(DOCX)

## Acknowledgments

We acknowledge the contributions of Karen Johnson (McCormick Dementia Services) in supporting recruitment and conducting focus groups during earlier study phases.

## Author contributions

**Conceptualization:** Kelly Kay, Victoria Smye.

**Data curation:** Arlene Astell.

**Investigation:** Kelly Kay, Kateryna Metersky, Rebecca H. Correia, Arlene Astell, Colleen McGrath, Winnie Sun, Halyna Yurkiv, Victoria Smye.

**Methodology:** Kelly Kay, Rebecca H. Correia.

**Project administration:** Rebecca H. Correia.

**Writing – original draft:** Kelly Kay, Rebecca H. Correia.

**Writing – review & editing:** Kateryna Metersky, Arlene Astell, Colleen McGrath, Winnie Sun, Halyna Yurkiv, Victoria Smye.

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
