## [Decision Letter · Decision Letter 0]

23 Oct 2025

Dear Dr. Correia,

**Kindly revise based on the reviewers suggestions for further consideration.**

We look forward to receiving your revised manuscript.

Kind regards,

Yogesh Kumar Jain, PhD

Academic Editor

PLOS ONE

**Journal Requirements:**

1. When submitting your revision, we need you to address these additional requirements. Please ensure that your manuscript meets PLOS ONE's style requirements, including those for file naming. The PLOS ONE style templates can be found at https://journals.plos.org/plosone/s/file?id=wjVg/PLOSOne_formatting_sample_main_body.pdf and https://journals.plos.org/plosone/s/file?id=ba62/PLOSOne_formatting_sample_title_authors_affiliations.pdf 2. In the online submission form, you indicated that “Anonymized survey data collected as part of the modified Delphi procedure can be made available upon reasonable request by contacting the corresponding author.” All PLOS journals now require all data underlying the findings described in their manuscript to be freely available to other researchers, either a. In a public repository, b. Within the manuscript itself, or c. Uploaded as supplementary information.This policy applies to all data except where public deposition would breach compliance with the protocol approved by your research ethics board. If your data cannot be made publicly available for ethical or legal reasons (e.g., public availability would compromise patient privacy), please explain your reasons on resubmission and your exemption request will be escalated for approval. 3. Your ethics statement should only appear in the Methods section of your manuscript. If your ethics statement is written in any section besides the Methods, please move it to the Methods section and delete it from any other section. Please ensure that your ethics statement is included in your manuscript, as the ethics statement entered into the online submission form will not be published alongside your manuscript. 4. We notice that your supplementary table is included in the manuscript file. Please remove and upload with the file type 'Supporting Information'. Please ensure that each Supporting Information file has a legend listed in the manuscript after the references list. 5. If the reviewer comments include a recommendation to cite specific previously published works, please review and evaluate these publications to determine whether they are relevant and should be cited. There is no requirement to cite these works unless the editor has indicated otherwise. 

Reviewers' comments:

**Comments to the Author**

1. Does the manuscript provide a valid rationale for the proposed study, with clearly identified and justified research questions?

Reviewer #1: Yes

2. Is the protocol technically sound and planned in a manner that will lead to a meaningful outcome and allow testing the stated hypotheses?

Reviewer #1: Yes

3. Is the methodology feasible and described in sufficient detail to allow the work to be replicable?

Reviewer #1: Yes

4. Have the authors described where all data underlying the findings will be made available when the study is complete?

Reviewer #1: No

5. Is the manuscript presented in an intelligible fashion and written in standard English?

Reviewer #1: Yes

You may also provide optional suggestions and comments to authors that they might find helpful in planning their study.

**Reviewer #1: ** This is an interesting, well-written paper describing a plan to develop a guide for dementia care delivery and education in Canada. The topic is very timely and pertinent for improving dementia care. The design appears sound, based on the modified Delphi method and published related studies.

One detail that could be explained better is how the 14 Principles would be distributed to the 80 panellists ("panellists will be assigned to three or four principles of dementia care", page 10, line 17). Would the panellists be able to choose which of the principles and hence statements they would be evaluating? Is there the potential for bias in that assignment? Some discussion of that issue would be helpful for the reader to better appreciate the plan.

Another detail that would strengthen the paper is a table listing the 14 principles. Even if they are presented in reference 13 (mentioned in the Discussion, page 14, line 9), the reader should not have to look up reference 13 for such an important element of the plan.

**Do you want your identity to be public for this peer review?** For information about this choice, including consent withdrawal, please see our Privacy Policy

Reviewer #1: No

---

## [Author Response · Author response to Decision Letter 1]

24 Oct 2025

Editor Comments to the Author

We confirm that we have reviewed these guidelines and that our manuscript complies with them.

2. In the online submission form, you indicated that “Anonymized survey data collected as part of the modified Delphi procedure can be made available upon reasonable request by contacting the corresponding author.” All PLOS journals now require all data underlying the findings described in their manuscript to be freely available to other researchers, either a. In a public repository, b. Within the manuscript itself, or c. Uploaded as supplementary information. This policy applies to all data except where public deposition would breach compliance with the protocol approved by your research ethics board. If your data cannot be made publicly available for ethical or legal reasons (e.g., public availability would compromise patient privacy), please explain your reasons on resubmission and your exemption request will be escalated for approval.

As this manuscript pertains to a study protocol, there is no underlying data available at this point that we can make available to other researchers. The anonymized survey data collected as part of this Delphi study will only be available in the future once the work described in this protocol is completed. Thereby, at a later point, interested readers can contact the corresponding author to request the anonymized survey data.

Not applicable. Our ethics statement only appears in the Methods.

4. We notice that your supplementary table is included in the manuscript file. Please remove and upload with the file type 'Supporting Information'. Please ensure that each Supporting Information file has a legend listed in the manuscript after the references list.

Thanks for clarifying this requirement. We have revised accordingly.

Not applicable. The reviewer did not recommend that we cite any previously published literature.

We have reviewed our reference list and can ensure it is complete and correct. We do not cite any retracted papers.

Reviewer #1 Comments to the Author

This is an interesting, well-written paper describing a plan to develop a guide for dementia care delivery and education in Canada. The topic is very timely and pertinent for improving dementia care. The design appears sound, based on the modified Delphi method and published related studies.

7. One detail that could be explained better is how the 14 Principles would be distributed to the 80 panellists ("panellists will be assigned to three or four principles of dementia care", page 10, line 17). Would the panellists be able to choose which of the principles and hence statements they would be evaluating? Is there the potential for bias in that assignment? Some discussion of that issue would be helpful for the reader to better appreciate the plan.

Thank you for the opportunity to clarify how participants will be assigned to the subset of principles to assess. As shown in the newly added Table 2, the 14 principles (and corresponding behavioural statements) have been organized into three work packages to balance the number of behavioural statements per package (n=93 or n=94). Using information collected as part of the demographic survey, we will use these characteristics to balance the perspectives that panellists bring to the study and assign them evenly across the work packages. For example, there are equal proportions of individuals who are care partners versus health and social care providers assigned to each work package. Further, we made efforts to balance the disciplines of health and social care providers within each work package (e.g., relatively equal number of nurses versus social workers versus physicians versus allied health professionals within each work package).

Therefore, panellists do not choose which principles they are evaluating as we agree this would introduce bias. The revisions below in our main text now clarify these details:

Page 10: Given the large volume of competency statements to be assessed in this study (n=281), panellists will be assigned to three or four principles of dementia care out of 14 total (organized into “work packages”), and will only evaluate the associated behavioural statements within those principles (Table 2) [13].

Page 15: However, the large volume of statements (n=281) cannot be feasibly assessed by every panellist, given concerns about participant burden and survey fatigue, resulting in the decision to assign panellists to rate a subset of competency statements (n=93 or n=94; see Table 2). Efforts will be made to balance the characteristics, experiences, and qualifications of those assigned to each work package based on responses to the demographic survey (e.g., the proportion of care partners to health and social care providers; disciplines of providers).

8. Another detail that would strengthen the paper is a table listing the 14 principles. Even if they are presented in reference 13 (mentioned in the Discussion, page 14, line 9), the reader should not have to look up reference 13 for such an important element of the plan.

Thank you for this suggestion. We have added Table 2 in the manuscript to list the 14 principles and clarify the number of behavioural statements associated with each.

---

## [Decision Letter · Decision Letter 1]

29 Oct 2025

Establishing a dementia care competency framework for care partners, health and social care providers: A modified Delphi study protocol

PONE-D-25-46326R1

Dear Dr. Correia,

We’re pleased to inform you that your manuscript has been judged scientifically suitable for publication and will be formally accepted for publication once it meets all outstanding technical requirements.

Kind regards,

Yogesh Kumar Jain, PhD

Academic Editor

PLOS ONE

Additional Editor Comments (optional):

Reviewers' comments:

Reviewer's Responses to Questions

**Comments to the Author**

1. Does the manuscript provide a valid rationale for the proposed study, with clearly identified and justified research questions?

Reviewer #1: Yes

2. Is the protocol technically sound and planned in a manner that will lead to a meaningful outcome and allow testing the stated hypotheses?

Reviewer #1: Yes

3. Is the methodology feasible and described in sufficient detail to allow the work to be replicable?

Reviewer #1: Yes

4. Have the authors described where all data underlying the findings will be made available when the study is complete?

Reviewer #1: No

5. Is the manuscript presented in an intelligible fashion and written in standard English?

Reviewer #1: Yes

You may also provide optional suggestions and comments to authors that they might find helpful in planning their study.

Reviewer #1: The addition of Table 2 is a solid improvement. My concerns regarding the first draft have been satisfactorily addressed.

**Do you want your identity to be public for this peer review?** For information about this choice, including consent withdrawal, please see our Privacy Policy

Reviewer #1: No

---

## [Editor Report · Acceptance letter]

PONE-D-25-46326R1

PLOS ONE

Dear Dr. Correia,

I'm pleased to inform you that your manuscript has been deemed suitable for publication in PLOS ONE. Congratulations! Your manuscript is now being handed over to our production team.

Kind regards,

on behalf of

Dr. Yogesh Kumar Jain

Academic Editor

PLOS ONE